# Multiparametric Dual-Time-Point [^18^F]FDG PET/MRI for Lymph Node Staging in Patients with Untreated FIGO I/II Cervical Carcinoma

**DOI:** 10.3390/jcm11174943

**Published:** 2022-08-23

**Authors:** Matthias Weissinger, Stefan Kommoss, Johann Jacoby, Stephan Ursprung, Ferdinand Seith, Sascha Hoffmann, Konstantin Nikolaou, Sara Yvonne Brucker, Christian La Fougère, Helmut Dittmann

**Affiliations:** 1Department of Nuclear Medicine and Clinical Molecular Imaging, University Hospital Tuebingen, 72076 Tuebingen, Germany; 2Department of Diagnostic and Interventional Radiology, Eberhard-Karls-University Tuebingen, Hoppe Seyler-Straße 3, 72076 Tuebingen, Germany; 3Department of Diagnostic and Interventional Radiology, University Hospital Tuebingen, 72076 Tuebingen, Germany; 4Department of Women’s Health, University Hospital Tuebingen, 72076 Tuebingen, Germany; 5Institute for Clinical Epidemiology and Applied Biometry, University Hospital Tuebingen, 72076 Tuebingen, Germany; 6iFIT-Cluster of Excellence, Eberhard Karls University Tuebingen, 72074 Tuebingen, Germany; 7German Cancer Consortium (DKTK), Partner Site Tuebingen, 69120 Heidelberg, Germany

**Keywords:** [^18^F]FDG PET/MRI, multiparametric imaging, dual-time-point kinetic, cervical carcinoma, lymph node metastases

## Abstract

[^18^F]FDG PET/MRI was shown to have limited sensitivity for N-staging in FIGO I/II cervical carcinoma. Therefore, this prospective study aimed to investigate the additional value of multiparametric dual-time-point PET/MRI and to assess potential influencing factors for lymph node metastasis (LNM) detection. A total of 63 patients underwent whole-body dual-time-point [^18^F]FDG PET/MRI 60 + 90 min p.i., and 251 LN were evaluated visually, quantified multiparametrically, and correlated with histology. Grading of the primary tumor (G2/G3) had a significant impact on visual detection (sens: 8.3%/31%). The best single parameter for LNM detection was SUVavg, however, with a significant loss of discriminatory power in G2 vs. G3 tumors (AUC: 0.673/0.901). The independent predictors SUVavg, ∆SUVpeak, LN sphericity, ADC, and histologic grade were included in the logistic-regression-based malignancy score (MS) for multiparametric analysis. Application of MS enhanced AUCs, especially in G2 tumors (AUC: G2:0.769; G3:0.877) and improved the accuracy for single LNM from 34.5% to 55.5% compared with the best univariate parameter SUVavg. Compared with visual analysis, the use of the malignancy score increased the overall sensitivity from 31.0% to 79.3% (Youden optimum) with a moderate decrease in specificity from 98.3% to 75.6%. These findings indicate that multiparametric evaluation of dual-time-point PET/MRI has the potential to improve accuracy compared with visual interpretation and enables sufficient N-staging also in G2 cervical carcinoma.

## 1. Introduction

Cervical cancer is the fourth most common cancer in women worldwide [1]. It affects young women starting in their 20s with the highest incidence at the age of 40 in the US and EU (of 15.1/100,000) [1,2]. Lymphatic spread occurs frequently already in early-stage cervical cancer, which mostly presents with (micro-) metastases [3,4]. These small metastases are hard to detect by CT or MRI, but the presence of lymph node metastases (LNM) is the most important prognostic factor in early tumor stages [2,3,4,5,6,7], and decisions on primary treatment (surgery vs. radiochemotherapy) depend on nodal involvement.

As cervical carcinoma is staged using the clinical FIGO classification, systematic lymphadenectomy, despite being associated with high morbidity, is still the gold standard for N-staging [2,7,8,9]. To reduce morbidity, sentinel lymph node (SLN) biopsy was introduced in 1999 for cervical carcinomas, proving to be safe for early-stage cancer in the case of successful SLN labeling [2,7,10,11,12,13]. However, even with correct tracer injection, SLN mapping can fail owing to strong venous tracer outflow or a transformation of the tumoral lymphatic drainage in pre-existing lymphatic tumor spread [14,15,16]. In addition, parametrial infiltration, while increasing the risk of LNM from 1% to 5–20% and fundamentally changing clinical management, often remains undetected until surgery [17,18]. Furthermore, the SLN technique was reported to be insufficient for the evaluation of the para-aortic LN status [16,19].

As a consequence, efforts have been made in recent years towards enabling more accurate and noninvasive N-staging by means of new imaging techniques, contrast agents, and tracers [20]. In this context, MRI reaches a very high specificity of about 95%, but only unsatisfying sensitivity of about 50% in early tumor stages [21]. However, the combination of MRI and [^18^F]FDG PET ([^18^F]FDG PET/MRI) improves the diagnostic accuracy in detecting pelvic and para-aortic LNM as well as distant metastases significantly [22]. Nevertheless, the sensitivity of the visual assessment of [^18^F]FDG PET/MRI in cervical carcinoma, even by experts, is limited owing to the low tumor-to-background ratio, especially of small LNM [16]. As histological ultrastaging revealed a much higher prevalence of isolated tumor cells in the LN or micro-LNM in early tumor stages than hitherto expected [23], the performance of [^18^F]FDG PET/MRI has to be improved.

Therefore, this study aimed to analyze the additional value of multiparametric PET/MR imaging comprising a dual-time-point [^18^F]FDG PET/MRI for N-staging in early tumor stages compared with expert reading using a swift, clinically applicable imaging protocol.

## 2. Materials and Methods

This prospective trial was registered in the German Clinical Trial Register (DRKS-ID: DRKS00014346) and approved by the institutional review board (registry no. 173/2015BO01) [24]. All participants provided written informed consent. A total of 69 consecutive patients with histopathologically confirmed cervical carcinoma and clinically determined stage ≤ FIGO IIB underwent whole-body dual-time-point [^18^F]FDG PET/MRI. A total of 63 of 69 participants underwent preoperative SLN mapping with SPECT/CT, followed by intraoperative SLN detection with a gamma probe and surgical staging between March 2016 and October 2020, and were included in the analysis (Figure 1).

### 2.1. PET/MRI Protocol

All patients underwent whole-body dual-time-point PET/MR after injection of about 3 MBq [^18^F]FDG per kg body weight (150–250 MBq, Biograph mMR^®^, (Siemens Healthineers, Erlangen, Germany), axial field of view: 258 mm, 4 × 4 × 20 mm LSO crystals, sensitivity: 14.1 cps/kBq, full width at half maximum @1 cm: 4.6 mm, no time-of-flight). Patients were asked to fast for at least 8 h, and blood sugar levels had to be below 150 mg/dl at injection. The early PET/MRI scan started with the first pass from midthigh to skull base 64.7 ± 11.7 min p.i., immediately followed by the delayed scan covering the inguinal and pelvic LN levels 90.6 ± 12.6 min p.i. An MRI contrast agent (8 mL Gadovist^®^) was applied except when contraindicated. Detailed MRI parameters are presented in Appendix A.

PET and MRI acquisitions were performed simultaneously, and the images were fused for further analysis. Acquisition time was defined by the MRI sequences and was BMI-adapted at 4–6 min/bed position for the first scan and 12–16 min/bed position for the delayed pelvic scan. Imaging data were reconstructed applying an iterative ordered subset expectation maximization algorithm (256 × 256 matrix) with a 4 mm Gaussian filter. Attenuation correction was performed using an MRI-based µ-map (SyngoMR E11^®^, Siemens Numaris/4, Siemens Healthineers, Erlangen, Germany).

### 2.2. SLN SPECT/CT

LN mapping was performed 3–5 h after intracervical injection of ≈ 200 MBq [^99m^Tc]Tc-Nanocolloide at the 3, 6, 9, and 12 o’clock positions. Imaging was performed with a hybrid SPECT/CT scanner (Discovery 670 Pro^®^, GE Healthcare, Chicago, IL, USA), as described previously [16]. An SLN was defined as focal activity enrichment in SPECT in a plausible anatomical region.

### 2.3. Histological Validation

Histological validation of LN was performed by removing the SLN, followed by a systematic pelvic lymphadenectomy. Para-aortic LN were removed if intraoperatively conspicuous or malignant SLN. [^99m^Tc]Tc-Nanocolloide-labeled SLN were localized and identified intraoperatively through a laparoscopic gamma probe (Neoprobe^®^, Models 1017 and 1100, Devicor Medical Products, Inc., Cincinnati, OH, USA), resected separately, and sent for frozen sectioning. SLN were further ultrastaged with the preparation of the entire LN in 200 µm slices.

### 2.4. Image Evaluation and Data Quantification

The evaluation of PET/MRI images with malignancy assessment of LN was performed prospectively in consensus by one radiology and one nuclear medicine specialist each with at least 8 years of experience in PET and MRI imaging. Anatomical positions of resected LN were identified on PET/MRI images based on their position in SLN SPECT/CT and the surgeon’s description of the localization intraoperatively.

Multiparametric data were collected using a dedicated software (syngo.via^®^ 8.2; Siemens Healthineers) and matched retrospectively with histology. Volumes of interest (VOI) were placed manually around every histologically confirmed LN on early and (standardized uptake value = SUV_e_) and delayed PET (SUV_d_). Quantification was performed as SUVmax and SUVpeak as well as SUVmean (50% isocontour). Blood pool correction (bpc) for SUV measurements (bpcSUV) was performed by dividing the lesions SUV by the SUVmean (without isocontour) of an ROI placed in a large venous vessel in the same PET bed position.
(1)$$\mathrm{bpcSUV}=\frac{\mathrm{SUV}\;\mathrm{VOI}}{\mathrm{SUV}\;\mathrm{blood}\;\mathrm{pool}}$$

Dual-time-point [^18^F]FDG kinetics were calculated using a retention index (RI), as described by Nogami et al. [25], and extended with a blood pool correction.
(2)$$\mathrm{RI}=\frac{\mathrm{bpcSUVd}-\mathrm{bpcSUVe}}{\mathrm{bpcSUVe}\;\times 100\%}$$

In addition, the absolute difference of the bpcSUV between the early and delayed scan was defined as SUV∆.
(3)ΔSUV =bpcSUVd−bpcSUVe

LN diameters were measured in the perpendicular short and long axis in the transaxial plane. Sphericity was defined as the ratio of short- to long-axes diameter. Diffusion was quantified manually using an ROI in the apparent diffusion coefficient (ADC) maps in LN ≥ 4 mm.

### 2.5. Statistical Analysis

Statistical analysis was performed using the SPSS Statistics 25.0 software (IBM Inc., Armonk, NY, USA), MedCalc v20.009 (MedCalc Software Ltd., Ostend, Belgium), and R 4.0.3 (R Foundation for Statistical Computing, Vienna, Austria). All parameters acquired were benchmarked against the gold standard histology. Differences in prevalence were tested for significance using the Chi² test. Differences between the means of groups were analyzed using the two-tailed *t*-test.

Listwise deletion was performed in case of missing values. Optimal cut-off values in ROC analyses were set at the Youden optimum.

The newly defined malignancy score (MS) predicts the probability of a lymph node exhibiting malignant histology based on a mixed logistic regression model, including the multiparametric imaging measures. This model uses the optimally weighted combination given the included predictors and covariances in the sample predicting the histological findings and incorporates random intercepts for patients within which the individual nodes account for dependencies. The probabilities are predicted for the current sample without using these random effects as these will not be known in future cases or samples for which one may wish to use the procedure. The criterion for statistical significance was set at α = 0.05.

## 3. Results

### 3.1. Patient Cohort

In total, 251 LN from 63 patients were assessed histologically and quantified multiparametrically with [^18^F]FDG PET/MRI. A total of 211 of 251 LN were located within the FOV of the delayed scan, enabling dual-time-point [^18^F]FDG kinetic calculation. A total of 219 of 251 LN had a sufficient size for ADC calculation. A total of 79 of 251 LN from 54 of 63 patients met the criteria for SLN in [^99m^Tc]Tc-Nanocolloide SPECT/CT. Detailed patient characteristics are presented in Table 1.

### 3.2. Prevalence of LNM Dependent on Stage and Grade of Primary Tumors

In 2 patients and 6 LN, respectively, no grading of the primary tumor was reported owing to conizations performed at other centers and no tumor was left when performing the (radical) hysterectomy in our center. The prevalence of LNM increased with the T-stage of the primary tumor (Figure 1). The patient-based prevalence of LNM was not significantly higher in patients with G3 (40%) than G2 (29.6%) tumors (*p* = 0.35). No LNM occurred in patients with G1 tumors.

### 3.3. Interrelationships of Histology and PET/MRI Parameters

LNM demonstrated a higher SUV, larger diameters, higher RI, and ∆SUV than benign LN, as detailed in Appendix A.

Moreover, these differences were amplified by the grade of the primary tumor, as presented in Figure 2 and Appendix A. In particular, LNM from G3 tumors presented with significantly higher SUV and FDG dynamics between early and delayed scan measured with RI-SUVavg (*p* = 0.03) and ∆SUVavg (*p* = 0.02) compared with LNM from G2 tumors (*p* < 0.01; Appendix A). Furthermore, G3 LNM presented with a greater short-axis diameter vs. G2 LNM (*p* < 0.01) and a slight increase in sphericity (*p* = 0.08), while ADC revealed no significant difference.

LN short-axis diameter correlated significantly with SUV_e_, SUV_d_, _BPC_SUV_e_, _BPC_SUV_d_, and ∆SUVpeak (*p* < 0.01, r: 0.477–0.716) but not with RI-SUVpeak (r = 0.085) or ADC (r = 0.241).

G3 LNM revealed an increase in [^18^F]FDG uptake between early and delayed scans compared with benign LN (RI-SUVpeak and ∆SUVpeak: *p* < 0.01 and 0.02), as presented for representative cases in Figure 3a,b. A similar trend was observed for RI-SUVpeak in G2 LNM, though not reaching significance (*p* = 0.19).

### 3.4. PET/MRI Parameter Evaluation

PET demonstrated high accuracy in differentiating between LNM and benign LN using an SUV-based quantification with an AUC of up to 0.809 (Figure 4 and Appendix A) without significant differences between the SUV quantification parameters SUV_e_max, SUV_e_peak, and SUV_e_mean (*p* ≥ 0.54).

The delayed PET scan did not result in a significantly higher AUC than the early PET scan (*p* ≥ 0.55). Blood pool correction improved the AUC in the delayed PET slightly but nonsignificantly (SUV_e_avg: 0.784 vs. 0.766; SUV_d_avg: 0.741 vs. 0.767, *p* = 0.73).

The primary tumor grade crucially impacted the accuracy of LNM detection in PET with a significant decrease in discriminatory power in G2 versus G3 tumors (SUV_e_avg G2: 0.673; G3: 0.901, *p* < 0.01). The error rate (ER = false-positive + false-negative rate = 1-accuracy) was more than twice as high for G2 LNM (65.5%) as for G3 LNM (30.4%) at their individual optimal SUV_e_avg cut-off (Appendix A), while the prevalence was comparable (G2: 17.5% vs. G3: 23.0%).

Dual-time-point kinetics calculated with RI and ∆SUV significantly correlated with malignancy, especially in G3 tumors with an AUC up to 0.791 (*p* < 0.01). The SUVpeak quantification method achieved the highest AUCs but required blood pool correction. Overall, the ∆SUV calculation method was comparable to the RI-SUV but performed slightly and nonsignificantly better in G3 tumors (G3 SUVavg: 0.791 vs. 0.718, *p* = 0.48).

LN diameters revealed a significant discriminatory power for short-axis (0.741) and long-axis (0.777) measurements and performed best in LNM from G3 tumors (AUC: 0.904 and 0.881). LN sphericity was not a significant stand-alone predictor of LNM, neither in G2 nor G3 tumors (*p* ≥ 0.269).

ADC presented a borderline significant discriminatory power (AUC 0.600, *p* = 0.05), with a significantly lower AUC compared with the SUVavg and short-axis diameter (*p* < 0.01 and *p* = 0.03, *n* = 162).

### 3.5. Multiparametric Approach

The parameters ADC, sphericity, bpcSUV_e_avg, and tumor grade of the primary tumor were identified as independent predictors of LNM and were included in the calculation of the MS, as described above. The response variable of the model were the probabilities of being malignant predicted by the model, calculated as a sum of the predictor values weighted according to their (fixed effect) regression coefficients. After listwise exclusion of cases with missing parameters, the sample size was 171 LN with 21.1% prevalence of metastases.

Using MS resulted in a high discriminatory power between malignant and benign LN (AUC: 0.820, 95% CI: 0.736–0.879). At the optimal cut-off value (Youden optimum: 0.042), the MS improved sensitivity from 63.5% to 72.2% compared with SUV_e_avg at a specificity of 80.7%.

Furthermore, error rates could be lowered (47.0%) and kept constant over a wider cut-off range compared with the best single parameter SUV_e_avg (52.7%), as presented in Appendix A.

Further subgroup analysis focusing on the grade of the primary tumor revealed a significantly (*p* < 0.01) better prediction of LNM in G3 tumors (AUC 0.850, 95% CI: 0.755–0.945) compared with G2 tumors (AUC 0.695, 95% CI: 0.526–0.863). In particular, the parameter SUV_e_ showed a markedly different predictivity for LNM in G2 compared with G3 tumors (log-odds: SUV: 1.5/17.7, *p* = 0.01).

### 3.6. Additional Value of Dual-Time-Point [^18^F]FDG Kinetic

The implementation of dual-time-point parameters significantly improved the model fit. The most predictive parameters were ∆SUVpeak (log-likelihood: −42.66; χ^2^ difference = 7.11; *p* < 0.01) and RI-SUVpeak (log-likelihood: −43.44; χ^2^ difference = 5.20; *p* = 0.02; log-likelihood of the comparison model without these dual-time-point parameters: −46.21, *n* = 144).

Implementing the dual-time-point [^18^F]FDG kinetic parameter ∆SUVpeak in the MS lowered error rates in G2 tumors by one-third from 65.5% to 44.5% compared with the best single parameter SUVavg (Appendix A).

Inclusion of ∆SUVpeak and RI-SUVpeak resulted in a slightly but not significantly increased discriminatory power (MS + ∆SUVpeak: AUC: 0.837; sensitivity: 79.3%; specificity: 75.7%) compared with the standard MS model (AUC: 0.820; sensitivity: 72.2%; specificity: 80.7%).

### 3.7. Visual vs. Multiparametric Evaluation

Specificity was set by the visual evaluation, and corresponding sensitivity was compared between visual and multiparametric LN evaluation using MS. Applying MS increased the overall sensitivity from 31.0% to 37.9% compared with the expert consensus at a set specificity of 98.3% (*n* = 144, prevalence: 20.1%), although the defined specificity was far from the Youden optimum of the MS (sensitivity: 79.3%; specificity: 75.6% at cut-off of 0.0042).

For G3 tumors, MS revealed a higher sensitivity (47.1% vs. 58.8%) compared with the human reader at a set specificity of 96.3% (*n* = 71, prevalence 23.9%), which was close to the Youden optimum (sensitivity of 76.4% at a specificity of 85.1%; cut-off: 0.0908).

For G2 LNM, using MS, resulted in an identical sensitivity of 8.3% at a set specificity of 100% (*n* = 73, prevalence: 16.4%). However, sensitivity increased from 8.3% to 83.3% if adjusted to the Youden optimum at a specificity of 72.1% (cut-off: 0.0435).

## 4. Discussion

To our knowledge, this is the first prospective study analyzing the additional diagnostic value of a multiparametric [^18^F]FDG PET/MRI analysis compared with expert consensus reading for N-staging with histology as the gold standard in FIGO I/II cervical carcinomas. A multiparametric malignancy score was introduced, which integrates dual-time-point [^18^F]FDG kinetics and biopsy-based grading of the primary tumor in addition to established PET and MRI parameters. Using [^99m^Tc]Tc-Nanocolloide for SLN labeling provided accurate transfer of LN positions via SLN SPECT/CT to PET/MRI, resulting in high data quality, which is a strength of this study.

Our results indicate that multiparametric analysis using the MS may double the sensitivity in LNM detection in FIGO I/II cervical cancer in G2 tumors compared with visual evaluation. As PET/MRI has already been shown to improve T- and M-staging, enhancing the accuracy in N-staging is the next big step in optimizing noninvasive staging for cervical carcinoma. This is of high clinical relevance, as surgical LN staging is currently the first step of surgery in advanced cervical cancer (in contrast to early cancer, where radical hysterectomy is usually the first step, followed by (sentinel-) LN dissection) [2]. Furthermore, preoperative assessment and evaluation of nodal involvement have a direct therapeutic impact as the presence of LNM leads to a change from radical hysterectomy to radiochemotherapy according to current guidelines [2].

### 4.1. Impact of Tumor Grade

Another key finding of this study was the strong influence of tumor grade on [^18^F]FDG uptake and [^18^F]FDG kinetics of LNM, which fundamentally affects their visibility in PET. As grading is usually assessed by biopsy as part of the initial gynecological examination, this is available when PET/MRI scan is performed.

The integration of histological characteristics into a multiparametric imaging-based analysis adds complementary information. In particular, primary tumor histology changed both the weighting of the individual parameters and their cut-off values in our study.

The present data indicate that the low sensitivity of [^18^F]FDG PET/MRI for G2 LNM might rather be due to smaller size, low SUV, and discreet [^18^F]FDG kinetics compared with their hitherto often assumed lower prevalence. In fact, LNM prevalence in G2 was not significantly different from that in G3 tumors using ultrastaging as the gold standard [26]. This finding is of high clinical relevance, as a solely visual assessment of [^18^F]FDG PET comes with insufficient sensitivity for N-staging in G2 tumors. Under consideration of early data, it can be hypothesized that the SLN technique may achieve more accurate N-staging than visual evaluation of [^18^F]FDG PET/MRI in early-stage G2 carcinoma [16,27]. This is even more important as, currently, in stage T1B1 and lower (i.e., early cervical cancer below 2 cm), LN dissection of only the SLN is currently considered state of the art [2]. Thus, preoperative knowledge of positive LN has a direct impact on the surgical procedure.

### 4.2. Dual-Time-Point [^18^F]FDG PET

In contrast to previous studies, we introduced a short-period dual-time-point imaging protocol using an interval of 30 min instead of 1–2.5 h [25,27,28], which enables continuous scanning without the need for repositioning patients. Even for this short time interval, a significant increase in tumoral [^18^F]FDG uptake was found—calculated as RI as proposed by Nogami et al. [25]—which was a significant independent predictor of malignancy. Furthermore, it was shown for the first time that the increase in [^18^F]FDG uptake over time was only significant for G3 LNM but not for G2 LNM. G2 LNM presented with a slight decrease in SUV analogous to the decline of blood pool activity, which might be explained by lower metabolic activity and tumor cell density [29].

Blood pool correction was crucial for dual-time-point dynamic measurements owing to a decreasing blood pool and increasing scatter correction artefacts caused by rising activity concentrations in the bladder. Although the dual-time-point kinetics were a significant factor, delayed PET images did not outperform the early scans. This might be due to LNM with increasing SUV dynamics already showing increased uptake on early PET scans.

### 4.3. Experts vs. Malignancy Score

Visual evaluation of LNM by expert readers was highly specific but accompanied by poor sensitivity, which runs counter to the principle of presurgical screening.

By using the MS with a cut-off value at the Youden optimum, the sensitivity could be improved substantially, especially in G2. The moderate loss of specificity would be acceptable, as false-positive pelvic LN are re-evaluated during surgery.

The sensitivity of visual evaluation was even lower than described in previous studies (31% vs. 45–88%) [30,31]. This might be due to our cohort of solely early-stage carcinomas and the higher number of micrometastases detectable by ultrastaging, as discussed above.

### 4.4. Limitations

The results presented here only pertain to G2 and G3 tumors as no LNM occurred in the small number of G1 tumors in our cohort.

The smaller FOV of the delayed [^18^F]FDG PET/MRI scan limited the dual-time-point analysis to pelvic LN.

In order to avoid further strain on the information extracted from the data, a listwise exclusion of cases was applied throughout the analyses; however, this resulted in a varying number of LNM in the results.

Furthermore, the multiparametric evaluation was based on histology and, therefore, was performed retrospectively in contrast to prospective reading of the experts. Consequently, the multiparametric analysis was subjected to accommodation of random effects to keep the diagnostic performance of MS comparable to the expert reading and other cohorts.

Prior to a broader clinical application, the presented MS should be validated prospectively in a comparable setting, which is planned for the second half of this ongoing clinical trial.

## 5. Conclusions

G2 vs. G3 tumor grade was identified as a crucial factor for limited visual detectability of LNM on [^18^F]FDG PET/MRI in early cervical carcinoma.

Multiparametric evaluation of dual-time-point [^18^F]FDG PET/MRI has the potential to considerably improve the accuracy of LNM detection and to extend sufficient N-staging also to G2 tumors.

## Figures and Tables

**Figure 1 jcm-11-04943-f001:**
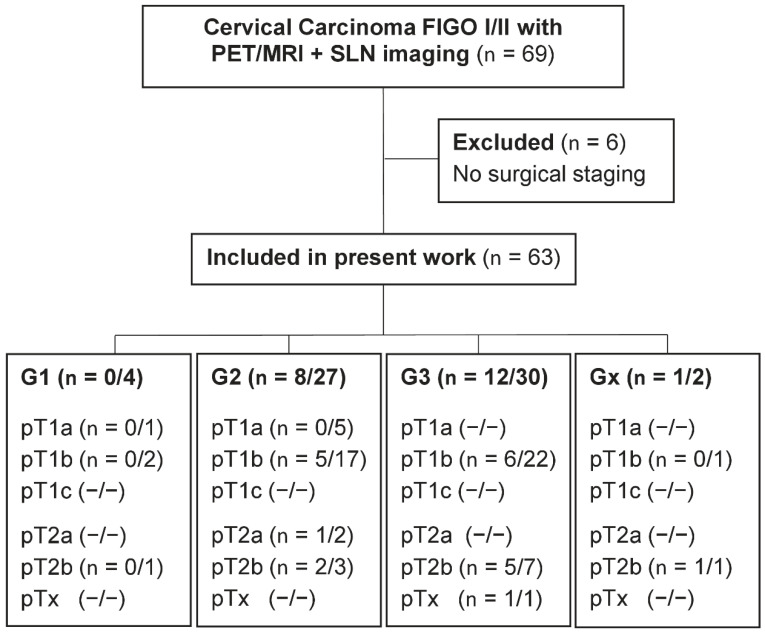
Consort flow diagram. Data are given as numbers of patients with LNM/all patients in subgroups.

**Figure 2 jcm-11-04943-f002:**
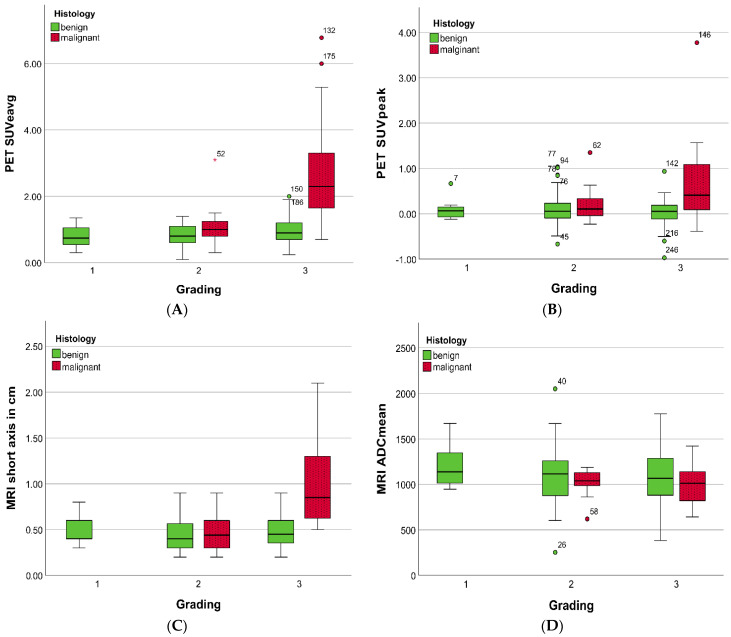
Boxplots presenting [^18^F]FDG PET (**A**,**B**) and MRI (**C**,**D**) parameters of lymph nodes dependent on the tumor grade of the primary tumor derived from biopsy before PET/MRI. No LNM were present in G1 carcinomas.

**Figure 3 jcm-11-04943-f003:**
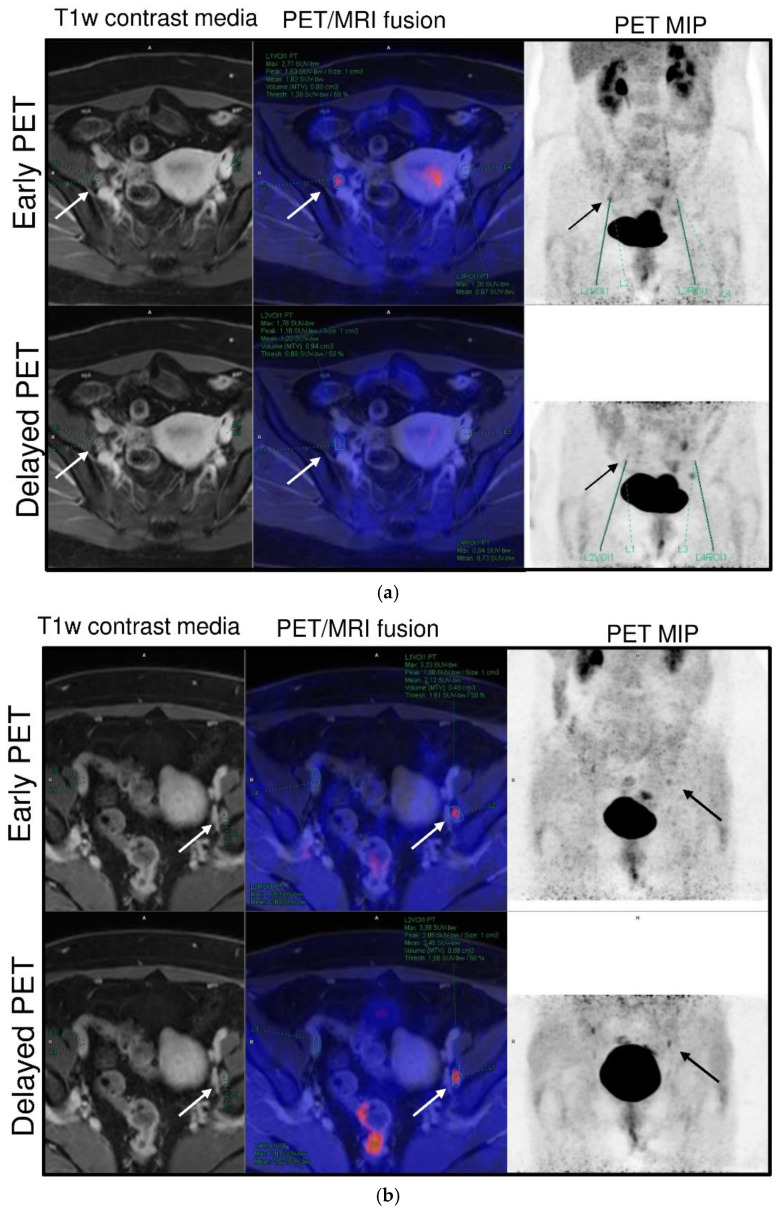
(**a**). Case of a 49-year-old patient with pT1b2 G3 cervical cancer. Focal [^18^F]FDG uptake (arrow) of the right interiliac LN decreased by 33% between early (60 min, SUVavg 1.8) and delayed PET scan (88 min, SUVavg 1.2) and was histologically confirmed as lymphofollicular hyperplasia. (**b**). Case of a 41-year-old patient with pT2b G3 cervical cancer. The left iliac extern LNM (arrow) presents an ongoing [^18^F]FDG trapping between the early (60 min, SUVavg 2.1) and delayed scan (82 min, SUVavg 2.5) and a slight decrease in blood pool activity.

**Figure 4 jcm-11-04943-f004:**
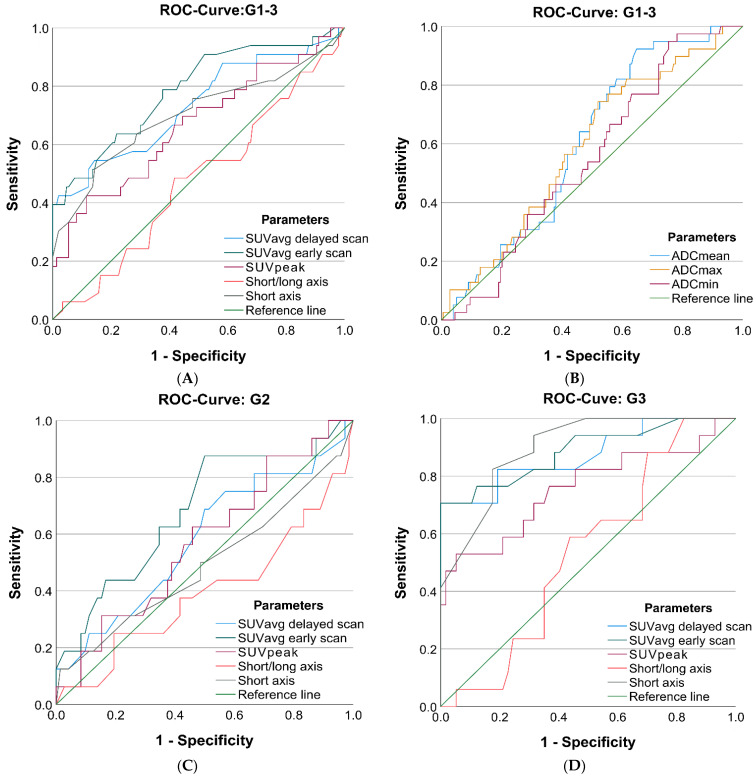
ROC analysis for the detection of lymph node metastases of selected [^18^F]FDG PET/MRI parameters for G 1-3 tumors (**A**) and (**B**) as well es G2 tumors (**C**) and G3 tumors (**D**) separately.

**Table 1 jcm-11-04943-t001:** Patient characteristics (*n* = 63).

	**Average** ± **SD**	**Range**
Age at PET/MR (years)	46.8 ± 11.5	28–72
Patient height (cm)	166 ± 6.6	152–187
Patient weight (kg)	71.0 ± 16.2	44–117
BMI (kg/m²)	25.7 ± 5.4	15–40
Time between PET/MR and LN histology (days)	22.4 ± 16.7	1–89 *

* One outlier with 89 d but no LNM at histology.

## Data Availability

The data presented in this study are available on request from the corresponding author. The data are not publicly available due to data protection regulations.

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
