# Peer review of "Multiparametric Dual-Time-Point [18F]FDG PET/MRI for Lymph Node Staging in Patients with Untreated FIGO I/II Cervical Carcinoma"

_jcm, 2022, doi:10.3390/jcm11174943_

Round 1
Reviewer 1 Report
- I would like to thank the authors for this very nice work.
- could the authors give some more information on the scanner that was used, for instance TOF capabilities, special resolution, detector technology, etc, since this is of great influence on the detection of small lesions.
Author Response
Dear Reviewer 1,
thank you very much for the fair revision of our manuscript. We fully agree with you that instrumentation and reconstruction can have a strong effect on SUVs and therefore thresholds can differ greatly between the scanners. We have included supplemental information on detectors, resolution, and TOV in the methodology as suggested. The Biograph PET/MR was the 2nd Siemens scanner worldwide and did not yet have the TOF function.
The following changes were made (see also in the track change version):
“(axial field-of-view: 258mm, 4x4x20mm LSO crystals, Sensitivity: 14.1cps/kBq, full width at half maximum @1cm: 4.6mm, no time-of-flight)” Page 6, line 130-132.
We hope that we have adequately responded to your comments and questions with our amendments.
Reviewer 2 Report
These finding indicate that multiparametric evaluation of dual-time-point PET/MRI has the potential to improve accuracy compared to visual interpretation and enables sufficient N-staging also in G2 tumor. However, some key questions have not been answered or supplemented with relevant data. Therefore, major revision is needed to further improve this manuscript.
1. The authors should better summarize the innovative nature of this manuscript.
2. The authors should provide more cases and perform statistical analyses.
3. The quality of the pictures is poor and should be improved.
4. Some literature on tumor imaging should be cited (e.g., Coordination Chemistry Reviews, 2020, 419:213393; Journal of Bio-X Research, 2020, 3(04): 144-156; Chemical Society Reviews, 2021, 50(15): 8669-8742; ACS Applied Materials & Interfaces, 2017, 9(48): 41782-41793; Military Medical Research, 2022, 9(1): 1-18.)
5. Minor grammatical errors should be avoided and authors should add language polish to the manuscript.
6. The format of references should be consistent.
Author Response
Dear Reviewer 2,
thank you very much for the detailed and critical review of our manuscript. We have discussed your comments together at length and consider that we have been able to satisfy your comments and issues by making the following amendments. In the following, we would like to provide the explanations and the changes for each point. The detailed changes can also be found in the track change version.
- The authors should better summarize the innovative nature of this manuscript.
-> Thank you very much for this comment. We have discussed this issue at length with the surgically active coauthors from the Women's Hospital and have added detailed amendments to the discussion regarding the clinical impact of an improved imaging prior surgery.
The following changes were made in the manuscript.
…” and decisions on primary treatment (surgery vs. radiochemotherapy) depend on nodal involvement”, Page 5, line 87
“This is of high clinical relevance, as surgical LN staging is currently the first step of surgery in advanced cervical cancer (in contrast to early cancer where radical hysterectomy is usually the first step, followed by (sentinel-) LN dissection) (2). Furthermore, preoperative assessment and evaluation of nodal involvement have a direct therapeutic impact as the presence of LNM leads to a change from radical hysterectomy to radiochemotherapy according to current guidelines (2).” Page 11+12 line 322-328
“This is even more important as currently in stage T1B1 and lower (i.e. early cervical cancer below 2 cm) LN dissection of only the SLN(s) is currently considered state-of-the-art (2). Thus preoperative knowledge of positive LNs has a direct impact on the surgical procedure.” Page 12, line 349-352
- The authors should provide more cases and perform statistical analyses.
-> More data and a collective to validate the results would be desirable. We want to realize this in a follow-up study. Unfortunately, this study has reached the number of subjects approved by the Ethics Committee, so that no further patients can be recruited. However, we are pleased to present the largest collective of patients with FIGO 1+2 cervical cancer studied with PET/MRI and SLN.
- The quality of the pictures is poor and should be improved.
-> Thank you for pointing this out. Unfortunately, there was apparently a loss of quality due to the reformatting. By a direct export from SPSS to .pdf with 300dpi we could ensure a very high image quality. All figures are now given with highest image quality in a separate pdf, the figures inserted in the word document are the image files with quality loss.
- Some literature on tumor imaging should be cited (e.g., Coordination Chemistry Reviews, 2020, 419:213393; Journal of Bio-X Research, 2020, 3(04): 144-156; Chemical Society Reviews, 2021, 50(15): 8669-8742; ACS Applied Materials & Interfaces, 2017, 9(48): 41782-41793; Military Medical Research, 2022, 9(1): 1-18.)
-> Thank you for the recommended Citations. We have added the appropriate citations to the manuscript.The following sentence was added to insert the proposed citation No. 20. (Wang XZ, Xiaoyanb; Lei, Hualia; Yang, Nailina; Gao, Xiangc; Cheng, Liang. Tumor microenvironment-responsive contrast agents for specific cancer imaging: a narrative review. Journal of Bio-X Research. December 2020;3:144-156)
“As a consequence, efforts have been made in recent years towards enabling more accurate and non-invasive N-staging by means of new imaging techniques, contrast agents and tracers (20).”
- Minor grammatical errors should be avoided and authors should add language polish to the manuscript.
-> Thank you very much for your comment. The manuscript has been linguistically polished by our coauthor Mr. Usprung with native speaker level English skills (PhD. degree from Cambridge) after the content adjustments.
- The format of references should be consistent.
Thank you for noticing formatting errors here. We have rechecked the bibliography created with EndNote for errors.
We hope that we have adequately responded to your comments and questions with our amendments.
Round 2
Reviewer 2 Report
The authors have made serious revisions and I recommend acceptance of this manuscript.